# An Enzyme- and Label-Free Fluorescence Aptasensor for Detection of Thrombin Based on Graphene Oxide and G-Quadruplex

**DOI:** 10.3390/s19204424

**Published:** 2019-10-12

**Authors:** Yani Wei, Luhui Wang, Yingying Zhang, Yafei Dong

**Affiliations:** 1College of Life Sciences, Shaanxi Normal University, Xi´an 710119, China; weiyani@snnu.edu.cn (Y.W.); wangluhui@snnu.edu.cn (L.W.); 2School of Computer Science, Shaanxi Normal University, Xi´an 710119, China; zhangyingying@snnu.edu.cn

**Keywords:** thrombin detection, fluorescence, aptamer, N-methyl-mesoporphyrin IX (NMM)

## Abstract

An enzyme- and label-free aptamer-based assay is described for the determination of thrombin. A DNA strand (S) consisting of two parts was designed, where the first (Sa) is the thrombin-binding aptamer and the second (Se) is a G-quadruplex. In the absence of thrombin, Sa is readily adsorbed by graphene oxide (GO), which has a preference for ss-DNA rather than for ds-DNA. Upon the addition of the N-methyl-mesoporphyrin IX (NMM), its fluorescence (with excitation/emission at 399/610 nm) is quenched by GO. In contrast, in the presence of thrombin, the aptamer will bind thrombin, and thus, be separated from GO. As a result, fluorescence will be enhanced. The increase is linear in the 0.37 nM to 50 µM thrombin concentration range, and the detection limit is 0.37 nM. The method is highly selective over other proteins, cost-effective, and simple. In our perception, it represents a universal detection scheme that may be applied to other targets according to the proper choice of the aptamer sequence and formation of a suitable aptamer-target pair.

## 1. Introduction

In the fields of biomedicine and biotechnology, it is essential to detect bioactive molecules with high sensitivity and selectivity. Therefore, the emergence of biosensors has provided a convenient and fast method for concentration determinations of biomolecules, such as proteins, genes, and ions. In past research, antibody-linked immunosorbent assays (ELISA) have been widely used to detect proteins [1,2]. However, the approach is expensive and takes a lot of time to complete, as it requires incubation and multi-step washing processes [3,4]. Aptamers are single-stranded nucleic acids, i.e., DNA or RNA, which may be selected from random sequence nucleic acid libraries by an in vitro selection process named SELEX (Systematic Evolution of Ligands by Exponential Enrichment) [5,6]. Compared with antibodies, aptamers have many advantages, such as simple modification, easy production, chemical stability and reusability, easy storage, and low cost [7,8,9,10,11,12]. Aptamers have great binding affinity and high specificity for a variety of targets, for example, small molecules, proteins, metal ions, or cell surfaces [13,14,15]. Therefore, aptamers are considered to be the promising candidate for the construction of sensors, including electrochemical sensors, fluorescence aptasensors, and colorimetric sensors [13,16,17].

Recently, fluorescent methods have attracted extensive attention in the field of sensing devices based on aptamers thanks to their high sensitivity and simple operation. According to previous strategies, we have observed that most aptasensors have been designed based on fluorophores and quenchers [18,19,20]. However, the modification of aptamers is expensive, time-consuming, and requires complex processes. In addition, modifications with fluorophores and quenchers affect sensitivity due to the weak interaction between aptamers and the corresponding targets [21]. Therefore, taking these constraints into account, a label-free method needs to be proposed for fluorescence detection of thrombin.

Thrombin, a multifunctional protease, plays a critical role in biology and pathological processes, such as thrombosis, cardiovascular disease, and early diagnoses and treatment [22]. In addition, thrombin is virtually nonexistent in the blood of healthy people, although low nanomolar concentrations of thrombin can be produced in early hemostatic processes [23]. Furthermore, high picomolar concentrations of thrombin can be detected in the blood of people with coagulation abnormalities, resulting in certain serious diseases, for example, thromboembolic disease and Alzheimer’s disease [24,25,26,27]. Therefore, given the great importance of detecting the concentration of thrombin in blood, it is necessary to propose an approach with excellent selectivity and specificity. In prior studies, many approaches for have been developed, such as label-free detection [28,29], fluorescence detection [13,28,29,30,31,32,33,34,35], etc. These methods are simple and easy to undertake, and have attracted considerable attention. Some researchers have introduced protease into the detection system, which is sensitive to the reaction conditions [27], and the modification of the aptamer undoubtedly affects the binding ability of the aptamer to the corresponding target [36]. As far as these methods are concerned, they are complicated and expensive. Overall, our method, an enzyme-free, non-label, universal fluorescence aptasensor is more capable of meeting the needs for the detection of thrombin and other targets.

As a new type of nanomaterial, graphene oxide (GO), comprises a single layer of carbon atoms in a densely-packed, honeycomb, two-dimensional lattice [37,38]. GO has many excellent properties, including good water dispersibility [36], high mechanical strength [39,40], etc. GO has a unique adsorption capacity, which can selectively absorb ss-DNA by π-π accumulation, instead of ds-DNA or G-quadruplex [41,42]. At the same time, GO can quench the fluorescence of dyes labeled with ss-DNA through the Förster resonance energy transfer (FRET) process [43]. Up to now, GO has been widely used to construct biosensors thanks to these excellent properties [44,45,46,47,48,49,50]. Nanomaterials have been applied to the recognition mechanisms of biomolecules, which has given rise potential applications in the fields of biomolecular detection systems [51,52]. In addition, as a special deoxyribonucleotide structure, G-quadruplex has also been introduced into our detection system. It is G-rich fragment that has a significant fluorescent signal after interaction with NMM [50,53]. G-quadruplex overcomes the shortcomings of expensive DNA modification. There is no doubt that G-quadruplex is the promising candidate as a new signal reporter. So far, G-quadruplex has attracted considerable attention in the construction of biosensors and molecular detection [54,55,56,57,58].

Here, through the combination of an advanced nanomaterial (GO) with a special deoxyribonucleotide structure(G-quadruplex), we constructed an enzyme-free, label-free, and simple approach for sensitive fluorescent detection of thrombin. In the prescience of thrombin, we obtained high fluorescence upon the formation of a duplex of the target/aptamer. This enzyme- and label-free method can successfully detect thrombin, which makes it simple to use, time-saving, and cost-effective. This method provides a universal platform which can detect targets specifically according to the specific binding of the aptamer to its target.

## 2. Experimental Section

### 2.1. Materials and Reagents

DNA oligonucleotides were purchased from Shanghai Sangon Biotechnology Co (Shanghai, China); their sequences are listed in Table 1. Thrombin was derived from bovine plasma (lyophilized powder, 40–300 NIH units/mg protein). Lysozyme, thrombin, lgG, human serum albumin (HSA), and bovine serum albumin (BSA) were purchased from Sigma (St. Louis, MO, USA). Trypsase was purchased from Shanghai Sangon Biotechnology Co. (Shanghai, China). N-methylmesoporphyrin IX (NMM) was purchased from J&K Scientific Ltd. (Beijing, China). The stock solution of NMM was diluted with dimethyl sulfoxide (DMSO) and stored at −20 ℃. Graphene Oxide (GO) was purchased from Nanjing XFNANO Materials Tech Co., Ltd (Nanjing, China) and suspended in water via sonication. All other materials were purchased from Xi’an JingBo Bio-Technique Co. (Xi’an, China). The fetal bovine serum samples were obtained from the Lanzhou Roya Bio-Technology Co. (Lanzhou, China), Led, and diluted with a reaction buffer 100 times before use.

### 2.2. Apparatus

Fluorescence emission spectra were measured using an EnSpire ELIASA from PerkinElmer (Waltham, MA, USA). The fluorescence spectra of NMM was excited at 399 nm and emitted at 610 nm. A fluorescence intensity at 608 nm was used to evaluate the performance of the experiments. 

### 2.3. The Sensing Procedure

First, thrombin, S (0.25 µM) and the buffer (20 mM Tris-HCl, 100 mM NaCl, 10 mM KCl, 10 mM MgCl_2_, pH = 7.5) were mixed for 30 min at 37 ℃. Then, GO (30 µg·mL^−1^) was added into the reaction system and mixed for 30 min at room temperature. Next, NMM (1.5 μM) was added to the final reaction samples, and the fluorescence intensity was measured after mixing for 25 min at 37 ℃. Finally, we recorded the fluorescence spectra.

### 2.4. Selectivity of the Thrombin Assay

S (0.25 µM) was mixed with thrombin (0.2 µM) or other non-specific proteins (Lysozyme, Trypsase, lgG, BSA and HSA 2 µM) respectively. Subsequent operations were consistent with the process described above.

## 3. Results and Discussion

### 3.1. Principle of the Method

The proposed method for label-free and enzyme-free thrombin detection based on GO and G-quadruplex is illustrated in Scheme 1. The well-designed DNA strand S was introduced into the system, which consisted of two parts: Sa is the thrombin aptamer sequence, and Se is the sequence of the G-quadruplex. First, S bound to the surface of the GO through π-π stacking; in the presence of thrombin, it binds to the Se part (aptamer sequence) due to the high binding affinity between the target and the corresponding aptamer. The ss-DNA could be anchored to the surface of the GO; the Se/Thrombin duplex would then strip the G-quadruplex from the GO. After that, in the final stage, NMM was inserted into the G-quadruplex and used to emit a high fluorescence. In the absence of thrombin, S was still anchored to the GO surface, resulting in a low fluorescence intensity. Consequently, thrombin could be detected simply and with sensitivity by observing the fluorescence intensity change of NMM. The proposed method could be used for the detection of other targets by changing the corresponding aptamer.

### 3.2. The Feasibility of the Strategy

We check the feasibility of thrombin detection by recording the fluorescence intensities under different conditions. As shown in Figure 1, only if the NMM was added individually could a weak fluorescence of NMM (curve e, green curve) be obtained. However, a significant fluorescence intensity enhancement (curve b, red curve) was recorded because of the interaction between the G-quadruplex and NMM. However, in the coexistence of S and GO, the system showed a low fluorescence of NMM after the incubation of S with GO for 30 min (curve d, purple curve), due to the absorption function between GO and ss-DNA(S). The fluorescent signal was regarded as a background signal, which proved the high quenching ability of GO. In contrast, when thrombin was present, the hybridization between thrombin and its aptamer promoted the formation of a duplex of thrombin/Sa, which liberated the S form GO and notably increased the NMM fluorescence (curve c, blue curve) compared with the curve d (in the absence of a target). These results confirm the feasibility of the proposed biosensor. 

### 3.3. Optimization of Experimental Conditions

To achieve the best performance, we optimized the various experimental conditions, including the concentration of S, the concentration of GO, the reaction time between the G-quadruplex and NMM, and the length of terminal signal carrier sequence. F and F_0_ denote the fluorescence intensity of NMM in the presence and absence of thrombin. F_0_ is regarded as a blank group.

In the method, G-quadruplex is the terminal signal carrier and has a potentially significant effect on the intensity of the fluorescence. The long length of the G-quadruplex had a weak interaction with GO, resulting in a high quenching efficiency and low background signal. As shown in Figure 2A, F/F_0_ is increased as the length of G-quadruplex is increased (from one-quarter complete to complete G-quadruplex). Finally, F/F_0_ reached a peak when the length of the G-quadruplex was 17-mer.

As shown in Figure 2B, the initial fluorescence was low with lower concentrations of S, and the change in the intensity of the fluorescence was not obvious. However, the system presented the opposite results with higher concentrations. The F/F_0_ gradually increased to a range of 0.15 μM to 0.25 μM, and reached a maximum at 0.25 μM, before subsequently decreasing. Therefore, due to its optimal signal-to-noise level, the best concentration of S was found to be 0.25 μM in follow-up experiments. According to Figure 2C, the ΔF value gradually increased with increasing the GO concentration in the range from 10 µg·mL^−1^ to 30 µg·mL^−1^, before gradually decrease thereafter. The mixed solution containing a low concentration of GO resulted in a high background signal; however, a high concentration of GO led to fluorescence quenching of the duplex of S/Thrombin/NMM. Furthermore, 30 µg·mL^−1^ was selected to achieve a better signal. In addition, the fluorescence intensity was also influenced by the interaction time between the G-quadruplex and NMM. As shown in Figure 2D, F/F_0_ has a high ratio of signal-to-noise (SNR), reaching a maximum at 25 min.

### 3.4. Analytical Performance

The evaluation criteria of the analytical performance are sensitive and play an important role in accurate detections. Based on optimized conditions, we evaluated the sensitivity by varying the concentration of thrombin. Figure 3A shows that a gradual increase in the fluorescence signal was attained as the concentration of thrombin increased from 0 µM to 50 μM. The fluorescence intensity at 608 nm is linear to the concentration of thrombin; the calibration plot obtained is shown in Figure 3B. The linear regression equation can be described as follows: Y = 1900.5 × Log_10_Concentration + 1467.6, with a good correlation coefficient of 0.9947. According to the 3б rule (3б/ S, in which б is the standard deviation for the blank solution and S is the slope of the linear equation), the limit of this detection method for thrombin is estimated to be 0.37 nM. Compared to other methods, the method proposed in the paper has a low detection limit (Table 2). More importantly, it is simple, due to its enzyme- and label-free design. 

### 3.5. Selectivity of Thrombin Detection

The specificity of the method was investigated with different, nonspecific proteins, including lysozyme, trypsase, lgG, human serum albumin (HSA), and bovine serum albumin (BSA) at a concentration of 2 µM to study the selectivity. As shown in Figure 4, the detection system emitted a weak fluorescence response to the nonspecific proteins. However, due to the specificity of the aptamer to its target, in the presence of thrombin, enhanced fluorescence of NMM was observed in comparison to other circumstances. These data demonstrate the excellent selectivity of the fluorescent aptasensor; it can be regarded as a specific and selective platform for target protein detection.

### 3.6. Determination of Thrombin in Actual Samples (Serum)

In order to verify the feasibility of the assay in real samples. Experiments were carried out by adding different concentrations of thrombin to bovine serum. A significantly enhanced fluorescence signal could be observed in both the buffer and serum compared with the blank control (Figure 5). In addition, we measured the recovery; the range was shown to be 97%–110% (Table 3). Overall, the results show the excellent applicability of the proposed method for thrombin detection in real samples.

## 4. Conclusions

In summary, this paper proposes a non-label and enzyme-free fluorescence aptasensor based on graphene oxide and G-quadruplex for thrombin detection. In our work, the aptamer could bind to its target, and thus, be separated from GO. After adding NMM, it emitted strong fluorescence. Additionally, the system showed excellent selectivity in the detection of other proteins. Our methodology has several distinct advantages. First, an aptamer probe without any label is involved, which is critical for retaining the excellent binding activity between the target and its aptamer, and makes the process cost effective, time-saving, and simple. Second, the feasibility and applicability of the mechanism were verified theoretically and empirically. Third, as a new type of nanomaterial, GO is easy to prepare in large quantities; it is a cost-effective and universal platform which can meet the needs of future practical applications. Moreover, our platform can be applied to the detection of other targets by changing the aptamers to form a suitable aptamer-target pair. Other molecules also can be detected, such as acetamiprid, cocaine, etc. In other words, our assay has potential applications in the fields of biology, biomedicine, biochemistry, and so on.

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
