# Peer review of "An Enzyme- and Label-Free Fluorescence Aptasensor for Detection of Thrombin Based on Graphene Oxide and G-Quadruplex"

_sensors, 2019, doi:10.3390/s19204424_

Round 1

Reviewer 1 Report

Dear author,

This is a nice manuscript and it describes a optionally useful assay. However, I do have some remarks:

the English needs improvement; in the introduction you mention the ELISA as being expensive,unstable and time consuming; however, this is not true and in practice the ELISA takes as much time as your assay, it is easy to perform and rather stable; in the evaluation of your assay there are some points missing: you should discuss the following points - working range; detection limit; specificity/selectivity; matrix effects; recovery; intra- and inter-assay variation.Further, you should discuss the applicability with regards to your findings. Finally, you should compare your results with those found in literature.

Reviewer 2 Report

The paper would describe a novel approach for thrombin determination.

The novelty of the approach is unclear and any critical comparison with standard thrombin approaches can be found in the paper. Moreover, the analytical results are in my opinion disputable as clearly described below.

For these reasons, I’m sorry to suggest AT LEAST reconsideration after major revision.

Introduction section. Aptamers, graphene oxide and so on have been well introduced and explained in this section. Anyway, it is not clear at all which is the novelty of the approach. Is the approach original or other papers already described a similar approach? Furthermore, and not less important, the Authors didn’t describe any analytical approaches already used to determine thrombin in clinical sample, so any comparison can be done. Optimization of experimental conditions, page 5. Here the Authors optimised several analytical variables but didn’t explain any about. For example, Fig 2 reports the influence of aptamer length and concentration but any rationale about the observed behaviour can be found. As the Authors well know, scientific papers are not cooking receipts while giving reasons for the observed behaviours can increase the quality of scientific report. Analytical performance, page 6. It is not clear at all why the emitted fluorescence is proportional to log of analyte concentration, instead to be proportional to the analyte concentration, as usually it is expected. Please explain the motivation for. Further, from a close inspection of calibration plot (Fig. 3.B) I’m not sincerely sure that a linear plot is observed and that the linear fitting is the best approach to fit the experimental points. R= 0.9834 in fact is not really a good correlation coefficient (as statistics states) and the distribution of points around the fitting line is clearly biased. Selectivity of thrombin detection, page 7. Firstly, it is not clear if the Authors used trypsin or trypsase: please correct. Furthermore, it is not clear why the Authors choose that compounds and that concentration levels to check selectivity: please explain. Finally, I’m unsure about the conclusions: as Fig. 4 shows, lysozyme interference, for example, was at least the 30 % of the thrombin signal so that the conclusion about the excellent selectivity and specificity (page 7) appear unreal. Determination of thrombin in actual sample, page 3.6 Firstly, the figure is unclear in my opinion: indeed, it is not clear to what the two different bars refers: please make clear the experiment and the relevant figure. More important, to test the analytical success of a new analytical procedure, it is necessary to compare with reference analytical approaches. Editing of the paper is required.

Round 2

Reviewer 2 Report

While some points have been addressed by the Authors, in the revised paper, some others, namely the most relevant, still remain unaddressed and quite unclear.

For these reasons, I’m sorry to suggest AT LEAST reconsideration after major revision.

Reply to Response 1. The Reviewer never doubted about the originality of the approach: just observed that this was not clear in the paper. Unfortunately, this remains still unclear in the revised paper since any statement about the novelty of the approach was described. Please, make it clear in the relevant introduction section. Response 2. This now addressed the point raised by Reviewer, so it is acceptable. Reply to Response 3. Reasons here described in the revised paper are poor in my opinion and not supported by any reference. Accordingly, response 3 poorly addressed the Reviewer points so I leave the decision to the Editor about the point. Reply to Response 4. This was not a rationale about the point raised by Reviewer but just data processing. As it is well known, fluorescence is proportional to analyte concentration, so it is perfectly unclear why the relationship was logarithmical. Please, give a rationale about or stress this unusual behaviour in the paper. Further, a logarithmical relationship with analyte concentration in undesirable since sensitivity of the analytical approach in not the same at every analyte concentration and gave troubles about the dynamic range of the analytical method. Please, stress also this undesired drawback in the paper describing the analytical performance of the proposed method. Reply to Response 5. As the Authors affirmed in their reply, repeating the calibration plot gave different results. This is a clear evidence of somewhat uncertain in the analytical method, particularly in the calibration plot, which is of paramount importance for analyte determination. Of course, the scientist cannot take the last points just because the correlation coefficient is better (or vice versa), but a thoroughly statistical study is required. Please, address this point. Reply to Response 6. Selectivity and specificity o f the proposed method cannot be tested just by using the first compounds in the lab, sorry. This is not a reason for choosing the substance tested in the paper. Please check for proteins or whatever that may be good candidate for aptamer binding. Reply to Response 7. The replies of the Authors here are unclear and do not address the point raised by Reviewer, i.e. the large interference from lysozyme and so the poor selectivity of the proposed method. Responses 8 and 9: the relevant points have been addresses now.
